# Hydrogels Classification According to the Physical or Chemical Interactions and as Stimuli-Sensitive Materials

**DOI:** 10.3390/gels7040182

**Published:** 2021-10-25

**Authors:** Moises Bustamante-Torres, David Romero-Fierro, Belén Arcentales-Vera, Kenia Palomino, Héctor Magaña, Emilio Bucio

**Affiliations:** 1Departamento de Biología, Escuela de Ciencias Biológicas e Ingeniería, Universidad de Investigación de Tecnología Experimental Yachay, Urcuquí 100650, Ecuador; 2Departamento de Química de Radiaciones y Radioquímica, Instituto de Ciencias Nucleares, Universidad Nacional Autónoma de México, Ciudad de México 04510, Mexico; david.romero@yachaytech.edu.ec; 3Departamento de Química, Escuela de Ciencias Química e Ingeniería, Universidad de Investigación de Tecnología Experimental Yachay, Urcuquí 100650, Ecuador; maria.arcentales@yachaytech.edu.ec; 4Facultad de Ciencias Químicas e Ingeniería, Universidad Autónoma de Baja California, Calzada Universidad 14418, Parque Industrial Internacional Tijuana, Tijuana 22390, Mexico; kenia.palomino@uabc.edu.mx

**Keywords:** hydrogels, physical interactions, chemical interaction, smart hydrogel, biomedical applications

## Abstract

Hydrogels are attractive biomaterials with favorable characteristics due to their water uptake capacity. However, hydrogel properties are determined by the cross-linking degree and nature, the tacticity, and the crystallinity of the polymer. These biomaterials can be sorted out according to the internal structure and by their response to external factors. In this case, the internal interaction can be reversible when the internal chains are led by physicochemical interactions. These physical hydrogels can be synthesized through several techniques such as crystallization, amphiphilic copolymers, charge interactions, hydrogen bonds, stereo-complexing, and protein interactions. In contrast, the internal interaction can be irreversible through covalent cross-linking. Synthesized hydrogels by chemical interactions present a high cross-linking density and are employed using graft copolymerization, reactive functional groups, and enzymatic methods. Moreover, specific smart hydrogels have also been denoted by their external response, pH, temperature, electric, light, and enzyme. This review deeply details the type of hydrogel, either the internal structure or the external response. Furthermore, we detail some of the main applications of these hydrogels in the biomedicine field, such as drug delivery systems, scaffolds for tissue engineering, actuators, biosensors, and many other applications.

## 1. Introduction

Novel biomaterials have been developed in the last decades. One of the most exciting for researchers is one of the first biomaterials, known as hydrogel [1]. The first hydrogel described arose in 1960, where the cross-linking network was based on polyhydroxyethylmethacrylate (pHEMA) [2,3,4]. A hydrogel is a tridimensional polymeric structure with swelling and collapse properties, flexibility, biodegradability, biocompatibility, and softness [5]. The ability of hydrogels to absorb water arises from hydrophilic functional groups attached to the polymeric backbone, while their resistance to dissolution arises from cross-links between network chains [6]. Variations in concentrations, structure, functionality of the monomer, and the cross-linker used in such gels can modify the structure [4]. Besides, these biocompatible materials have been widely used in the biomedical field due to their high ability to absorb drugs [7] and nanoparticles [8].

In hydrogel synthesis, monomers or polymers and an initiator are usually necessary. The latter will be responsible for the formation of monomeric free radicals that give rise to the growth of macromolecular chains. Monomers have unique properties that can form a macromolecular known as a polymer. In addition, a cross-linking agent is relevant because a characteristic of any hydrogel is its cross-linked structure, which is achieved through that agent [9,10]. The cohesion forces that allow the cross-linking of the polymer have a covalent character, and other forces such as electrostatic, hydrophobic, dipole–dipole interactions, or hydrogen bonds intervene [11,12,13]. Typically, hydrogels are classified into two categories: physically cross-linked hydrogels and chemically cross-linked hydrogels [14], which obtain different properties based on each synthesis.

In recent years, many efforts have been made on research approaches based on the synthesis of new hydrogels with better mechanical properties, which is one of these systems’ weak points, and provides them with a certain degree of “intelligence”. Since they are systems in an aqueous medium, it is necessary to consider the traditional variables such as temperature, concentration, pH, and ionic strength. Moreover, some hydrogels are capable of responding to external stimuli such as pH, temperature, electricity, light and biological molecules as enzymes during the swelling and shrinking process. Therefore, several studies have made it possible to improve mechanical, optical, or swelling behavior, adding another compound with hydrophobic properties to the hydrophilic monomer.

Hydrogels are composed of hydrophilic polymers, which are desirable materials in polymer science. They have different properties that make them potentially useful in a wide variety of applications, such as in biomedical applications, self-assembly, or catalysis [15]. Hydrophilic polymers might be considered as those polymers that contain polar functional groups such as hydroxyl (-OH), carboxyl (-COOH), and amino (-NH_2_) groups that make them soluble or swelled by water. A hydrophilic polymer that has received much attention is poly(vinyl alcohol) (PVA) because it has a great promise as a biological drug-delivery matrix [16] and is nontoxic. Similarly, hydrophilic polymers can be cross-linked through chemical bonds, leading to the formation of hydrogels, which are materials that have attracted particular attention in the biomedical field [17]. The research of hydrophilic polymers has been complex because the physical properties of solubility or swellability depend on different factors, such as the type of polymer, molecular weight, the ratio of polar groups, and degree of cross-linking. High molecular weight and a high degree of cross-linking will reduce the hydrophilicity of the molecule [18,19].

Hydrogels are attractive materials owing to their excellent features and properties. Besides, because of such a wide variety of response triggers, hydrogels can serve as sensors or actuators or can be utilized in controlled drug delivery systems, biosensors, tissue engineering scaffolds, and others [20], because of their biomimetic properties and multi functionalities [21].

## 2. Cross-Linking Strategies to Obtain Physical Hydrogels

Hydrogels can be cross-linked through physical or reversible networks, which hold them together by molecular entanglements or physicochemical interactions such as hydrogen bonds, hydrophobic interactions, charge condensation, or supramolecular chemistry. The interactions that occur in this type of hydrogels are weak. However, they are numerous and contribute to the presence of complex behaviors. Since the interactions depend significantly on external stimuli (pH, ionic strength, the composition of the solvent, or the temperature), they allow hydrogels to be highly versatile concerning the environment, unlike covalently bonded materials. Some of the gist hydrogels made up by the physicochemical are explained below.

### 2.1. Crystallization

Physical cross-linking of a polymer to form a hydrogel can also be achieved by crystallization through freeze–thaw cycles in homopolymeric systems or by forming stereocomplexes. The crystallization and degree of crystallinity determine the final properties of the resulting polymers. Polymeric crystallization can occur from dilute solutions or the molten state [22]. In the first case, crystallization occurs by evaporating the solvent, resulting in the appearance of single crystals based on a chain-folded model, which presents aligned chains [23]. This crystallization process includes crystal nucleation and crystal growth:

Crystal nucleation: The initial stage consists of the formation of tiny crystals or nuclei. It requires a certain degree of supersaturation, which increases the driving force for splitting the solution into a low and a high entropy region [24]. Several factors affect the crystal nucleation, such as temperature, volume, ionic strength, ionic species ratio, flow rate, and foreign particles [25].

Growth: It is characterized by the growth of the nuclei due to the alignment of molecular chain segments while part of the initial volume disappears. Polymers crystallize with chain-folded layers known as lamellae, which can be grouped forming spherical structures called spherulites. Polymers present crystalline phases when the lamellar chains are arranged in regular patterns and the amorphous phase when the lamellae are arranged irregularly. Generally, the polymers will crystallize from the melt in spherulitic structures [26]. The diameter of the spherulites is dependent on the nucleation sites, the molecular structure of the polymer, and the rate of cooling [27].

The crystallinity degree (%) measures the order in the molecular arrangement of polymers. It is calculated according to the following formula:(1)$$Crystallinity\;\left(\%\right)=\frac{\rho_{c}\left(\rho_{s}-\rho_{a}\right)}{\rho_{s}\left(\rho_{c}-\rho_{a}\right)}\times 100$$
where, $\rho_{c}$ is the density of the completely crystalline polymer, $\rho_{a}$ represents the density of the completely amorphous polymer, and $\rho_{s}$ is the density of the sample. The degree of crystallinity is dependent on the cooling rate and structure of the polymer [27]. It can range from a completely amorphous polymer (close to 0%) to a semicrystalline polymer (approximately 95%).

Freeze–thaw crystallization involves the formation of microcrystals in the structure. Examples of this type of cross-linking are xanthan [10,28] or PVA hydrogels [29,30]. The formation of resilient and resilient PVA gels is attributed to the formation of PVA crystallites that act as physical cross-linking sites in the network. The hydrogel properties depend on the concentration and molecular weight of the PVA, temperature, freezing time, and some freeze–thaw cycles [29]. It has been studied that by adding alginate to the PVA solution before subjecting it to the freeze–thaw process, the properties of the system can be modified (by increasing the concentration of alginate, the mechanical resistance increases, which causes a decrease in the drug release) [30].

Another form of crystallization is the formation of stereocomplexes [31]. Stereocomplexes are formed by stereoselective interactions between polymers with complementary stereoregular structures (the functional group is located only on one side of the monomer) that interact to form a system of properties different from those of the original constituents. It has been suggested that the forces involved in the formation of the complex are Van der Waals forces [32]. The formation and composition of the complexes between sterically complementary polymers are conditioned by medium composition or temperature [33].

For example, Ikada et al. [34] first described the ability of polylactic acid (PLA) to form stereocomplexes. Slager et al. [35,36] obtained microparticles by mixing D-PLA and L-PLA with leuprolide and observed that its release depends on the method of complex formation, on the molecular weight of PLA, on the leuprodyla: polymer and D-PLA ratios: L-PLA and other additives. Another system was created by Bos et al. [37,38], which was linked to the dextran with oligomers of D-lactic acid and L-lactic acid, separately. The formation of this hydrogel occurs at room temperature, physiological pH, and in an aqueous environment; the formation is not instantaneous, which allows its injection and gelation in situ. The system was shown to be biocompatible, biodegradable, and a releasing system for recombinant human interleukin-2.

### 2.2. Amphiphilic Copolymers

Amphiphilic hydrogels containing both hydrophilic and hydrophobic units represent one of the major polymeric biomaterials [39], as Figure 1 shows. In other words, amphiphilic copolymers can aggregate in water to form micelles and hydrogels in which the hydrophobic segments of the polymer self-assembly [40]. Copolymers typically form hydrogels with fragments of different nature or modified copolymers. The latter can be formed by a water-soluble polymer to which hydrophobic fragments have been attached or hydrophobic chains modified by water-soluble fragments [31].

By the PEG biocompatibility and PLA biodegradability (or its copolymer with glycolic acid (PLGA)), the hydrogels formed by these copolymer blocks have been extensively investigated. Drug release can occur by passive diffusion and by degradation of the system. Multiple triblock polymer systems with hydrophobic segments in between have been proposed, for example, PEG-PLGA-PEG at low concentrations in water form micelles and high concentrations form thermoreversible hydrogels. The critical gelation concentration and the sol-gel transition temperature are highly dependent on the blocks’ molecular weights and composition. PEG-PLGA-PEG copolymers gel when there is a change from room temperature to 37 °C if the concentration is high enough. Cross-linking is thought to occur by hydrophobic interactions [41].

Another example is the copolymers of PLA and polyoxyethylene (PEO), PLA-PEO-PLA, formed by polymerization of L-lactide with PEG. It has been suggested that these hydrogels can retain hydrophilic drugs in the PEG phase and hydrophobic drugs in the PLA domains. Likewise, they can be applied by injection to administer specific proteins [42,43,44].

Copolymers containing fragments of different nature, such as composed of PEG and polybutylene terephthalate (a hydrophobic polyester) have also been investigated. Feijen et al. [45,46,47,48,49] studied that films or microspheres can be formed, and they tested the release of various proteins (lysozyme, bovine serum albumin). They also studied the hydrolytic degradation of polymers, concluding that as the percentage of polybutylene terephthalate increases, the degradation will be lower. Furthermore, modifying the water/polymer ratio during emulsification could control the release of proteins.

Polymers with hydrophobic domains can also crosslink in aqueous environments through reverse thermal gelling (sol-gel transition). Polymers or oligomers with this ability are known as gelling agents and are moderately hydrophobic [50,51,52]. Gelation occurs when a hydrophobic segment attaches to a hydrophilic polymer forming an amphiphilic polymer. Amphiphilic polymers are generally soluble in water at low temperatures; as the temperature increases, the hydrophobic domains add to minimize the hydrophobic surface area, reducing the amount of structured water around them [53]. Gelation temperature depends on the concentration of the polymer, the length of the hydrophobic block, and the chemical structure of the polymer. Some hydrophobic segments that can undergo reverse thermal gelation at temperatures close to physiological ones are PLGA, polypropylene oxide, poly(N-isopropyl acrylamide) (PNIPAAm), polypropylene fumarate, polycaprolactone, polyurethane, polyorganophosphazene [51].

Different polysaccharides (such as chitosan, dextran, pullulan, carboxymethyl curdlan) can also be modified by adding hydrophobic segments that self-assemble to form nanohydrogels. Sunamoto et al. [54,55,56] obtained cholesterol-modified pullulan nanoparticles. These 20–30 nm nanoparticles (nanohydrogels) can be loaded with different proteins such as α-chymotrypsin, bovine serum albumin, insulin [55]; or drugs such as adriamycin [57]. By covalently binding galactoside lactoside to pullulan, nanoparticles are obtained whose cellular target is RCA lectin (specific receptor for β-D-galactose).

Another example is chitosan glycol-modified with palmitoyl chains that assemble into unilamellar polymer vesicles in the presence of cholesterol [58]. These vesicles are biocompatible and hemocompatible and can encapsulate water-soluble drugs [59]. Chitosan has also been modified with various hydrophobic fragments to form pH or temperature-sensitive hydrogels; for example, D-L-lactic acid [60] and/or glycolic acid [61], polyacrylic acid (PAAc) [62], and PNIPAAm [63]. Other polymers such as carboxymethyl dextran have also been modified with poly(N-isopropyl acrylamide)-co-N,N-dimethyl acrylamide to form thermo-sensitive hydrogels [64]. Finally, another example along similar lines is the modification of carboxymethyl curdlan (polysaccharide with antitumor activity) with a sulfonylurea to form nanohydrogels by self-assembly [65].

### 2.3. Hydrogel Cross-Linking by Charge Interactions

Cross-linking (or de-crosslinking) can be achieved in situ by pH changes that cause ionization or protonation of ionic functional groups and cause gelation. Charge interactions can occur between a polymer and a small molecule or between two oppositely charged polymers to form a hydrogel. When a polyelectrolyte combines with a multivalent ion of opposite charge, a physical gel known as an ionotropic hydrogel is produced [51].

If two polyelectrolytes of opposite charges are mixed, they can gel or precipitate depending on their concentration, ionic strength, and pH of the solution; the product is called complex or polyionic. In pioneering work in the area, calcium alginate capsules were stabilized, coating them with an alginate-poly(L-lysine) coacervate complex [5].

Starch graft copolymers with neutralized acid monomers generate anionic starches that can gel through charge interactions. Prado et al. [66] reported the formation of a novel interpolyelectrolyte complex (IPEC) between cation-ized corn starch by introducing the 2-hydroxy-3 (N,N,N-trimethylammonium) propyl group and κ-type carrageenan as counterpolyanion. It should be noted that carrageenan is made up of galactose and/or anhydrogalactose units, sulfated or not, linked by alternating bonds. The κ-type consists of alternating galactose units with a sulfate group at carbon four and unsulfated anhydrogalactose units. Gao et al. developed a simple, nontoxic, water-based strategy to fabricate magnetic nanoparticles/hydrogels nanocomposites in which highly crystalline Fe_3_O_4_ nanoctahedra can be fabricated in situ within a negatively charged hydrogel matrix [67]. Besides, Katayama investigated electrostatic interactions in polyampholyte gels, which contain anions and cations in their skeleton [68]. They observed that they deflated at neutral pH (pH = 7), while at various pH values, both higher and lower, they swelled. The reason is that, at neutral pH, charges attract each other in such a way as to result in a deflation of the gel. Conversely, if one of the charges is neutralized and the other ionized, the gel swells. It is important to note that both van der Waals forces and hydrogen bonds cause collapse at low temperatures, while hydrophobic interactions cause the opposite effect. Electrostatic interactions can be attractive and repulsive, depending on the nature of the gel.

### 2.4. Interactions by Hydrogen Bonds

Hydrogen bonding interactions can be used to produce hydrogels in vitro by freeze–thaw cycles. An example is a project developed by You et al. in which they report a hydrogel with a hydrogen-bonding system consisting of weak hydrogen bonds between N,N-dimethylacrylamide (DMAA), and acrylic acid (AAc) and strong hydrogen bonds between 2-ureido-4 [H]-pyrimidinone units. The hydrogels have unique properties through optimization between the radii of the monomers and a balance of the interactions [69].

Yoshimura et al. prepared biodegradable hydrogels by a simple procedure: esterifying starch with succinic anhydride, using 4-dimethylaminopyridine as catalyst and dimethylsulfoxide or water as a solvent, followed by neutralization with NaOH, dialysis, and precipitation with methanol. These hydrogels were obtained in the absence of a cross-linker. The authors hypothesize that gelation occurred due to aggregation of polymer chains by regeneration of hydrogen bonds during dialysis [70].

### 2.5. Stereo-Complexing

Stereo-complexing refers to the interactions between polymeric chains, or small molecules, of the same chemical composition but different stereochemistry. Natural polymers can be cross-linked by stereo-complexing grafting. The grafting of L-lactide and D-lactide oligomers to dextran induces spontaneous gelation in water [51]. Dextran is a polysaccharide similar to amylopectin, consisting of highly branched glucose chains, whose predominant bond is α (1→6) with α (1→3) and α (1→4) branches. These hydrogels show excellent biocompatibility and biodegradability. They do not require organic solvents, chemical crosslinkers, or the formation of hydrophobic domains. However, the main drawback of stereo- complexation is the restricted polymer composition range that can be used; small changes in stoichiometry can weaken or eliminate the stereochemical interaction [51].

Some physical hydrogels have found interesting applications. For example, Mehyar et al. investigated two physical hydrogels (one based on hydrogen-bonded starch and the other based on calcium alginate, with ionic interactions) as possible substrates for antimicrobial agents. In these systems, diffusivity depended on the hydrogel-agent pair, being the highest with the following combinations: trisodium phosphate, starch hydrogel, and sodium acid chlorite-alginate hydrogel [71].

### 2.6. Protein Interactions

Cross-linking by protein interactions can be accomplished through the use of genetically engineered proteins or antigen–antibody interactions. By employing genetic engineering, a genetic code can be designed that originates peptide sequences with specific physicochemical properties, and even synthetic amino acids can be obtained [72]. Tirrell [73] and Cappello [74] were pioneers in this field.

Cappello et al. [75] created a high molecular weight protein-polymer based on a sequence of silk-like amino acids and elastin, in which the insoluble silk-like segments associate in sheets or strands linked by hydrogen bonds. A particular subset of these silk-elastin-type protein compositions, called ProLastins, gelled in physiological solution. The sol-gel transition can be controlled by modifying the temperature, the conditions of the solution, and the additives, which can prevent or promote the crystallization of the chain through hydrogen bonds.

Tirrell et al. [76] used recombinant DNA methods to create artificial proteins that undergo reversible gelation in response to changes in pH or temperature. Proteins are composed of terminal leucine zipper domains flanking a flexible central segment formed by a water-soluble polyelectrolyte. The formation of coiled aggregates of the terminal domains in near-neutral aqueous solutions and at room temperature triggers the formation of a three-dimensional polymeric network, in which the polyelectrolyte retains the solvent and prevents chain precipitation. Increasing the pH or temperature dissociates the terminal aggregates, causing the dissolution of the hydrogel. The mild pH and temperature conditions in which the hydrogel is formed suggest that these hydrogels could be applied in encapsulation or controlled drug and cell release.

Another cross-linking by protein interactions is that formed by antigen-antibody interactions. For example, Miyata et al. [77] proposed modifying a PAAm hydrogel to give it the ability to reversibly swell in a buffer solution in response to a specific antigen. The system was prepared by binding an antigen and the corresponding antibody to the polymer network; thus, antigen–antibody binding increases the cross-links of the PAAm hydrogel. The competitive binding of free antigen in the medium causes a change in the hydrogel volume due to the breaking of these non-covalent cross-links. Furthermore, it was shown that the hydrogel behaves with shape memory and that gradual changes in antigen concentration can induce pulsatile permeation through the lattice.

## 3. Cross-Linking Strategies to Obtain Chemical Hydrogels

Hydrogels are called chemical or permanent when they consist of covalently cross-linked networks. Similar to physical hydrogels, chemical hydrogels are not homogeneous. It generally contains regions of the high cross-linking density and low degree of swelling (clusters), dispersed in the regions of the low cross-linking density and high swelling index. The presence of these clusters is due to the hydrophobic aggregation of the cross-linking agent. In some cases, depending on the composition, solvent, temperature, and solids’ concentration during gelation, phase separation can occur with the formation of macropores [51].

### 3.1. Graft Copolymerization and Irradiation Crosslinking

Among the various techniques for grafting polymers, gamma radiation has been successfully developed to graft polymers onto polymeric materials [78]. This technique does not require initiators or additives that may be harmful and difficult to remove [79]. Figure 2 illustrates this technique, requiring no initiator or cross-linking agent, and can be used with virtually any vinyl monomer. Both the polymerization reaction and the cross-linking can be started at room temperature. This radiation acts as an initiator of the copolymerization process between the polymer matrix (the material) and the molecule to be grafted (monomer). This method allows radiation to act on the polymer matrix, inducing the formation of reactive sites that may interact with a molecule to be grafted, initiating a free radical polymerization process [80].

Ali and AlArifi synthesized a series of starch/methacrylic acid (MAAc) copolymers of different compositions [81]. Both copolymerization and cross-linking were induced using γ-rays. They investigated the effect of the preparation conditions on the gelation process and found that increasing the total concentration of the reaction mixture and/or the irradiation dose improves both the conversion and the degree of gelation. They obtained superabsorbent materials with a maximum swelling of 1200% at 2 h and were sensitive to changes in pH.

Some synthesized hydrogels by irradiation have been tested in the controlled release of drugs. Eid reported the production of starch/MAAc copolymeric hydrogels with N-Vinyl Pyrrolidone (NVP) by irradiation. The hydrogels were loaded with vitamin B12, and their release rate was pH-dependent [82]. Bustamante-Torres et al. succeeded in synthesizing hydrogels based on agar and AAc through a graft copolymerization using gamma radiation. Using ciprofloxacin and silver nanoparticles, they used a Cobalt 60 source to cross-link these monomers and obtain pH-sensitive hydrogels with applications in charge and controlled drug release. The hydrogels were loaded with antimicrobial compounds and then were evaluated against methicillin-resistant Staphylococcus aureus and Escherichia coli, showing outstanding results [83].

A well-studied hydrogel is PHEMA which is obtained by polymerizing 2-hydroxyethyl methacrylate with a suitable cross-linking agent, for example, ethylene glycol dimethacrylate [84]. Water-soluble polymers have also been derivatized with methacrylic groups by the method proposed by Edman et al. [85], examples of which are dextran [85], hydroxyethyl starch [86,87,88], poly-aspartamide [89,90,91], and PVA [92].

UV radiation-induced polymerization is also frequently used to prepare hydrogels [92,93,94,95]. With this type of polymerization, they can be formed in situ [96], molded structures and even photoreversible systems can be prepared that photodegrade on exposure to UV light such that the release of drugs is controlled [97].

The limitation of UV light polymerization is that the radicals formed can cause damage to the structure of drugs [98] and proteins [31]. Cells exposed to high-intensity UV radiation for a long time can have their metabolic activity affected [99]. In addition, caution should be exercised with both the type of photoinitiator and the dissolved solvent, as these can be released from the hydrogel.

### 3.2. Reactive Functional Groups

These are covalent reactions between the functional groups of the polymers (mainly -OH, -COOH, -NH_2_) that provide solubility to water-soluble polymers [99,100]. Covalent bonds between polymer chains are established by the reaction of functional groups that have complementary reactivity; typical reactions are the formation of Schiff bases [101], Michael-type additions [102], peptide bonds [103], and click-type reactions [104]. Figure 3 illustrates an example of these reactions in order to form a cross-linked hydrogel structure.

Polymers containing hydroxyl groups and amino groups can be cross-linked by molecules that contain aldehyde functional groups in their structure. From these methods, the formation of a Schiff’s base between an aldehyde and an amino group is the most widely used technique. One of the most studied aldehyde cross-linking molecules in this field is glutaraldehyde, which reacts with amino groups under mild conditions. Examples of this reaction for the formation of hydrogels are the cross-linking of proteins (albumin [105], collagen [106], and gelatin [107], among others) and the cross-linking of polysaccharides (chitosan [108,109], guar gum [110], sodium alginate [111], and carrageenan).

However, the disadvantage of glutaraldehyde is its toxicity, even at low concentrations, causing possible leaching in the body during matrix degradation that would inhibit cell growth [112]. Due to the toxicity of this compound, other bifunctional small molecules have been proposed for use as cross-linking agents.

Similarly, the cross-linking process can also be carried out using addition reactions when units of a cross-linking agent are added without loss of atoms; that is, the chemical composition of the resulting chains is equal to the sum of the chemical compositions of the polymer and the organic cross-linking molecule. Widely developed examples for this type of hydrogel are polysaccharides cross-linked with 1,6-hexamethylene diisocyanate [113], divinylsulfone [114], or 1,6-hexane dibromide [115], among others. Typically, these addition reactions are carried out in organic solvents to prevent water from reacting with the cross-linking agent. These agents are often toxic; thus, the hydrogel must be treated to remove unreacted cross-linking molecules. For this reason, the drugs or proteins to be encapsulated must be loaded later [114]. It implies that it is necessary to use matrices with a pore size greater than said proteins to load proteins such that the subsequent release is generally of order one and with a limited duration [31].

Another cross-linking between reactive groups is caused by condensation reactions between hydroxyl or amino groups with carboxyl groups (or derivatives), which are frequently used to synthesize polymers to obtain polyesters or polyamides, respectively.

One of the most used reagents to cross-link water-soluble polymers through amide links is N-(3-dimethylaminopropyl) -N’-ethylcarbodiimide, adding N-hydroxysuccinimide to reduce possible side reactions and have better control over the cross-linking density that occurs in the hydrogel. By using such compounds, biocompatible polymers such as gelatin have been cross-linked [116].

Other of the most commonly used condensation reactions in the field of covalent hydrogel synthesis are the Passerini and Ugi reactions. In the Passerini reaction, an isocyanide, an aldehyde (or ketone), and a carboxylic acid form an α-acyloxyamide. This multicomponent reaction makes it possible to obtain polymeric cross-links employing ester-type bonds functionalized in position α degradable at different ranges of temperature and pH depending on the nature of the substituents. Degradation time varies from 1 to 8 days. Similarly, the Ugi reaction allows the formation of an α-(acylamine) amide linkages, adding a primary amine to the Passerini reaction. In this case, the cross-linking obtained is of the amide-type, giving the hydrogel more excellent stability and consistency [31]. Crescezi et al. described the synthesis of hydrogels based on polysaccharides by using this condensation reaction [117,118].

### 3.3. Enzymatic Method

The main advantage of the enzymatic method is that the cross-linking of the hydrogel occurs under mild conditions without the need for the use of low molecular weight compounds (monomers, initiators, cross-linking agents), irradiation, or prior polymer functionalization to favor its cross-linking [31,119]. Enzymes often exhibit a high degree of substrate specificity, potentially avoiding side reactions during cross-linking. This advantage makes it possible to control and predict the cross-linking kinetics and control the overall cross-linking rate. For this reason, this method is suitable for in situ gelation systems [119,120]. Figure 4 illustrates this method by unionizing the enzyme and the available substrate, forming a covalent bond within the cross-linking structure.

An example of this technique is that proposed by Chen et al. [119], who studied the cross-linking of gelatin and chitosan in the presence of transglutaminase and tyrosinase. They demonstrated that transglutaminase catalyzes the formation of resistant and permanent gelatin hydrogels (thermally irreversible) without requiring chitosan, although its presence makes the reaction faster and the hydrogels stronger. Tyrosinase was also studied to catalyze the chitosan–gelatin reaction. Although in this case, the presence of chitosan is necessary for the formation of the hydrogel. These hydrogels can change consistency with temperature, such that they become stronger when cooled and weaken when heated above the melting temperature of gelatin. The resistance of hydrogels catalyzed by both enzymes can be modulated by modifying the composition of gelatin and chitosan.

Another example of this technique is that proposed by Sperinde et al. [121], in which transglutaminase was used in a solution of PEG (tetrahydroxyl PEG functionalized with glutamimyl groups) and polylysine-phenylalanine. This enzyme catalyzes the reaction between the γ-carboxyamide group of PEG and the epsilon-amine group of lysine to obtain an amide bond between the polymers. The properties of the system can be modulated by changing the proportions of PEG and lysine.

For comparative purposes, Table 1 shows the different benefits and disadvantages of the synthesis of hydrogels by physical and chemical methods.

## 4. Smart Hydrogels

Although the first studies related to hydrogels appeared in 1894, the word “smart” was introduced in 1948 by Kuhn and co-workers [122]. Unlike conventional hydrogel, smart or stimuli-responsive hydrogels provide more efficient and valuable properties to the system and enhance the application level. Smart polymers or stimuli-responsive polymers respond to external stimuli and are characterized by their stimuli-responsive behavior due to different functionalities [123]. Figure 5 illustrates this reversible behavior under the presence of external stimuli. The constant changes would be approached to smart polymers based on the stimuli or signs requirements. An essential feature of these smart polymers is that the macroscopical changes are reversible, i.e., these systems can recover their initial state when the sign or stimuli ends [124]. The swelling properties of these hydrogels have attracted the attention of researchers and technologists who have found widespread applications in drug delivery devices, separation processes, sensors, contact lens devices, and many other fields [125].

### 4.1. pH-Sensitive Hydrogels

These polymers contain, in their structure, acid groups (carboxylic or sulphonic) or basic groups (ammonium salts) [126]. A variety of pH-responsive, biocompatible polymers exhibiting anionic or cationic properties allow these hydrogel networks to be tailored or modified to exhibit desirable physicochemical properties appropriate for the application at hand [127,128]. pH-sensitive polymers named polyacids or polyanions, such as PAAc or poly(methacrylic acid) (PMAA), have in their structure a significant number of ionizable acid groups, such as carboxylic acid or sulfonic acid [129]. The acidic groups deprotonate at high pH, while the basic groups protonate at low pH [130].

A polymer is considered pH-responsive when it comprises pendant acidic or basic moieties capable of donating or accepting protons upon an environmental change in pH [123]. pH-responsive hydrogels are synthesized from pH-sensitive polymers possessing ionizable functional groups which either accept or release protons in response to external changes [128,131,132]. The association, dissociation, and binding of various ions to polymer chains cause hydrogel swelling in an aqueous solution [130]. Therefore, these internal changes as a response would be approached into different fields.

### 4.2. Temperature-Sensitive Hydrogels

Temperature-sensitive hydrogels consist of water-filled polymer networks that display a temperature-dependent degree of swelling, where they can present a positive response (gel material swells with the increasing temperature) or negative temperature response (material shrinks with increasing temperature) [133]. Thermo-sensitive polymers contain in their structure hydrophilic and hydrophobic portions. When these polymers are subjected to temperature changes, the interaction between hydrophilic and hydrophobic segments in the polymer with water molecules are modified by changing the solubility of the cross-linked network, causing the sol-gel phase transition [134]. This transition is vital during the synthesis of this kind of intelligent hydrogels because they become temperature-dependent. Based on the manners of phase transitions, they could be classified into polymers with a lower critical solution temperature (LCST)- and an upper critical solution temperature (UCST)-type phase behavior [135,136,137]. Below the LCST, the polymer used to be hydrated state, which means soluble, but above the LCST, it becomes hydrophobic and precipitates [138]. In other words, LCST-exhibiting blocks may impart sol-gel transition with warming, but the UCST-exhibiting polymers typically trigger the reverse transition, that is, gel to sol [139]. Polymers with LCST can form negatively thermo-sensitive hydrogels, while positively thermo-sensitive hydrogels increase their water-solubility as the temperature increases, possessing UCST [140]. For LCST polymers, as the temperature increases, it results in negative free energy of the system, making the water-polymer association unfavorable, facilitating the interaction between polymer chains and water molecules [141].

### 4.3. Electro-Sensitive Hydrogels

Electro-sensitive polymers are a class of intelligent polymers that present a response to an external electric field. The gist characteristics of hydrogel based on the electro-sensitive polymer are the swelling or deswelling capacity under an electric field. These polymers are classified into ionic and dielectric polymers. Ionic electro-sensitive polymers are also known as conductive polymers, which result from electric field-driven mobility of free ions to create a change in the local concentrations of the ions in solution or within the material [142]. These polymers store and release charge through redox processes; for example, when oxidation occurs, ions are transferred to the polymer; in contrast, when reduction occurs, the ions are released back into the solution [143]. In other words, these materials are activated by a chemical reaction, changing from an acid to an alkaline environment and causing the gel to become dense or swollen, respectively [144].

Conversely, dielectric electro-sensitive polymers are those in which the electrostatic (Coulombic) forces that develop between the electrodes cause deformation of the polymeric material [145]. They have great potential for applications in both sensors and actuators [146].

### 4.4. Photo-Sensitive Polymers

Photo-responsive polymers are special polymers that respond to light and dark conditions and thus give rise to reversible variations in their structure and conformation [147]. Photo-responsive polymers are obtained by a photo-responsive functional group (chromophore) incorporated into the polymer chain, and based on the chromophore used, the response can be reversible or irreversible [148]. Polymers containing azobenzenes and spiropyran as the chromophore are the most widely studied photo-responsive polymers [149]. The changes in polarity and hydrophilicity are achieved by either isomerization or ionization, where the dimerization is considered a further light-induced process of high notability for hydrogels and film polymers [150]. The photo-responsive polymer can be for drug delivery applications [151].

### 4.5. Enzyme-Sensitive Hydrogel

Enzyme-sensitive hydrogels are also known as bio-sensitive hydrogels. These hydrogels employ enzymes as a cross-linking agent. Besides, degradation or morphological changes in the hydrogels can be led by the enzymes processes. Different enzymes are used in enzyme-responsive materials where some of the most frequently used enzyme classes include proteases, kinases, phosphatases, and endonucleases [152]. Since enzymes are highly selective, materials can be programmed to respond to a specific enzyme by incorporating the specific substrate (or a substrate mimic) [153]. Therefore, hydrogels must have substrate or recognition elements available and accessible for the enzyme, leading to chemical reactions between enzymes and substrate, leading to changes in the material properties [154].

## 5. Applications of Smart Hydrogels

Smart hydrogels have become an exciting and promising material in various applications thanks to their ability to respond to different endogenous and exogenous stimuli that confer unique properties. The sensing properties, biocompatibility, and similar mechanical properties to soft tissues [155] enable smart hydrogels suitable for industrial and biomedical fields. The responsive properties of smart hydrogels have also found immense potential to be applied in other applications such as self-healing [156], energy-storage materials [157], sustain release fertilizer [158], wastewater treatment [159], and others (Figure 6). Such applications depend clearly on the dynamics of the hydrogel networks, which can be designed to respond to different physical, chemical, or biological stimuli. Table 1 summarizes the main applications of smart hydrogels based on their stimuli-responsiveness. The most popular current applications of these materials are reported in more detail in the following section.
gels-07-00182-t001_Table 1Table 1Overview of potential applications of smart hydrogels based on their responsiveness.Hydrogel TypePotential ApplicationReferencesTemperature-responsiveTissue engineering, drug delivery, imaging, wound dressing, sensors.[140,160,161]pH-responsiveDrug delivery, sensing, 3D cell culture, antibacterial, Wastewater treatment, drug delivery, tissue engineering.[162,163,164,165]Light-responsiveMicrofluidic devices, drug delivery, soft robotic, actuators.[166,167,168]Magnetic-responsiveCancer therapy, regenerative medicine, drug delivery.[169,170]Electro-responsiveActuators, drug delivery.[171,172]


### 5.1. Drug Delivery Systems (DDS)

Smart hydrogels are gaining importance over conventional hydrogels for an efficient drug delivery system because their properties can be modulated in response to environmental stimuli. In particular, the volume-changing behavior of swelling and non-swelling of smart hydrogels makes them ideal as a drug-releasing trigger upon environmental factors [173].

Hydrogels can be fabricated by varying design parameters such as size (macro, micro, and nano), shape (spherical and non-spherical), and composition, factors that determine their use for drug delivery [174]. Hydrogels have been researched in terms of drug delivery because they are able to hold, within the cross-linked matrix, a number of different substances [175], which commonly are drugs. The well-known characteristics of the hydrogel and the drug that will be loaded provide essential information about their interaction. Hydrogels can be delivered into the human body by different routes such as surgical implantation, local needle injection, or systemic delivery via intravenous infusion [176]. Moreover, the releasing of drugs from hydrogels depends on the desirable outcome, and it can be achieved by different mechanisms such as swelling/deswelling, diffusion, and chemical mechanism.

Several approaches have been developed to produce innovative drug carrier systems that encapsulate drugs and deliver them to a specific site with a controlled release depending on environmental factors such as temperature, pH, glucose, light, ionic strength, and electric fields [177,178]. Temperature and pH have been the most currently investigated [179].

Typical examples of thermoresponsive polymers used for drug delivery include natural polymers such as chitosan/β-glycerophosphate (CS/GP), cellulose, gelatin, xyloglucan, dextran, and synthetic polymers such as poly(propylene glycol) (PPG), poly(ethylene glycol) (PEG), and PNIPAAm, the latter being one of the most studied [180,181]. Generally, polymers undergoing reversible phase transitions because of their lower critical solution temperature (LCST) are employed to develop potential drug delivery systems [182]. For example, PNIPAAm hydrogel is a promising drug delivery material due to its LCST around 37 °C. PNIPAAm can easily encapsulate drugs at temperatures below the LCST, whereas when heated to a temperature above the LCST, the hydrogels undergo a reduction in volume and release the drug. Despite their advantages in drug delivery, PNIPAAm hydrogels are non-biodegradable, limiting their use in clinical applications [183]. Therefore, to solve this problem, several strategies have been proposed to prepare hybrid PNIPAAm-based hydrogel with biodegradable species [184]. For instance, alginate-g-PNIPAAm hydrogels were prepared to evaluate the thermo-responsive release of doxorubicin (DOX), a hydrophobic anti-cancer drug. An in vitro test was performed, which showed a sustained release of the drug released from the hydrogels. Furthermore, effective drug delivery was achieved, with a better cellular uptake of DOX released from the hydrogel for overcoming cancer multidrug resistance [185].

Concerning pH-sensitive hydrogels, both acidic and basic polymers can be utilized. The nature of these types of hydrogels depends on the pH of the medium. For example, there is a pH variation in the human body along the digestive tract, from pH 2 in the stomach to 8 in the intestine [186]. Some of the most widely reported cationic polymers are chitosan and poly (ethylene imine). This type of hydrogel is characterized to swell at low pH due to the protonation of amino/imine groups [187]. By contrast, anionic polymers swell at higher pH because of the ionization of the acidic groups; some examples are carboxymethyl chitosan and PMAA. Han et al. prepared a capsule with a double-layer structure comprising polyacrylamide (PAAm) as a cationic layer and PAAc as anionic layer hydrogel study the lipophilic drug delivery. The releases simulation showed the potential for releasing the encapsulated drug (carotenoid) in a particular basic or acidic environment. Moreover, an in vitro test was performed, confirming this drug release method [188].

Hydrogels with other smart-responsiveness have also been investigated in several applications, such as glucose-responsiveness as potential insulin carriers [189] and magnetic-sensitive hydrogels as suitable drug delivery systems for cancer chemotherapy [177]. Moreover, significant efforts have also been focused on developing and designing dual-responsive hydrogels with enhanced properties [190,191]. Hoang et al. have recently synthesized a pH/thermo-responsive hydrogel comprising PAAc and norbornene-functionalized chitosan (CsNb) for colon targeted drug delivery. The results showed the hydrogel swelling and an almost complete drug release (92%) at the intestinal environment of pH 7.4 and 37 °C, whereas these activities were inhibited for an acid environment of pH 2.2. The obtained hydrogel exhibits stimuli-responsive properties that make it an excellent material for colon-targeted controlled drug release [192].

Despite a large number of research performed related to sensitive hydrogel-based drug delivery, only a small number of DDS are approved for clinical use by the Food and Drug Administration (FDA) [193,194]. Some of the facing barriers for their clinic use involve manufacturing cost, regulatory guidelines, and practical adaptability [195].

### 5.2. Scaffolds for Tissue Engineering

Recently, hydrogels with stimulus-responsive ability have attracted much attention in tissue engineering for the preparation of scaffolds. Scaffolds are three-dimensional porous materials designed for repairing or regenerating pathologically altered tissues, as illustrated in Figure 7. The biodegradability of scaffolds is an essential factor since scaffolds act as a temporary matrix for cells regeneration, which must be degraded during or after the healing process [196]. Materials to be employed as scaffolds must meet more certain required properties, which vary depending on the desired scaffold application and particular environment into which the scaffold will be implanted [186].

In this context, intelligent hydrogels are materials with novel features for preparing scaffolds owing to their biocompatibility, mechanical strength, porosity, and soft tissue-like properties [187]. Smart hydrogels have been studied to regenerate a wide range of tissues such as cartilage, bone, tendon, muscle, cornea, etc. [197]. To design scaffolds, both natural and synthetic smart hydrogels can be employed. Some of the natural polymer-based hydrogels include collagen, gelatin, and chitosan. In general, these polymers exhibited good properties in tissue engineering resulting from their natural and intrinsic characteristic of biological recognition. Moreover, examples of synthetic polymer-based hydrogels generally used in tissue engineering are PEGDA: poly(ethylene glycol) diacrylate, PLA: poly(lactic acid), PEG, and PEO. An advantage of synthetic polymers over naturally sourced is greater control of characteristics and properties for tissue regeneration [198].

Different approaches to develop scaffolds from smart hydrogels have been investigated. For example, thermal responsive hydrogels comprising alginate, gelatin, and different SiO_2_ NPs concentration, were developed as scaffolds to regenerate damaged tissues. The morphology analysis showed proper scaffold use for cartilage regeneration because their structure exhibits pores in the 88–207 mm range. Hydrogels containing SiO_2_ performed more excellent stability during in vitro biodegradation experiments in comparison to hydrogels without NPs. Moreover, the presence of SiO_2_ NPs supports cell growth and viability [199]. García et al. reported that a pH-responsive and antimicrobial hydrogel design resulted from the combination of difunctional polyethylene glycol dimethacrylates (PEGMA) and AAc as a monofunctional monomer. The ability of the hydrogel prepared by additive manufacturing to swell and shrink with pH changes was also reported. Several experiments were carried out for evaluating the matrix as a scaffold towards mammalian cells. Extensive washings with ethanol were necessary to remove possible unreacted monomers (acrylates and methacrylates) cytotoxic. Moreover, these washing were beneficial for improving the metabolic activity to close values to the ones found in tissue culture plates. Finally, the reported hydrogels showed excellent antimicrobial properties against *Staphylococcus aureus* [200].

### 5.3. Actuators

The development of actuators is considered one of the most exploited applications of smart hydrogels for diverse areas such as biosensing systems, targeted drug delivery, regenerative medicine, cell microenvironment engineering, and related fields. Stimuli-responsive actuators can convert chemical or physical energy (e.g., light, heat, pH, and magnetic field) into mechanical motion. Under stimulus cues, the swelling/deswelling process gives rise to materials capable of mimicking living systems such as motion and actuating functions [197,201]. The choice of the type of smart-hydrogel used is an essential step for the design of the actuators since its performance will depend on the mechanisms of response to external signals [202]. Moreover, owing to their biocompatibility, smart hydrogels can be used safely in bio-applications.

Several approaches describe the design and applications of hydrogel-based actuators with different stimuli-responsive properties. For instance, PNIPAAm has been studied as a potential candidate as a bioactuator because of its sensitivity to temperature changes [203]. Stoychev and coworkers developed partially biodegradable stark-like bilayers comprising PNIPAAm and hydrophobic polycaprolactone (PCL) layers. The PNIPAAm hydrogel exhibited swelling/deswelling behavior with temperature variation, whereas the PCL layer restricted the PNIPAAm movement. The resulting bilayer experienced folding and unfolding due to PNIPAAm swelling and collapse, respectively. Moreover, the whole bilayer structure evidenced its ability to control the capture and release of cells [204].

In order to enhance the performance of the hydrogel-based actuators, different types of additives, including zero-dimensional (0D), one-dimensional (1D), or two-dimensional (2D), are embedded into the hydrogel. Among 0D additives, several nanoparticles and metallic ions have been used. The integration of magnetic nanoparticles (such as Fe_3_O_4_ nanoparticles, Co nanoparticles, NdFeB microparticles, among others) into nonmagnetic responsive hydrogels leads to a material with a magnetically actuated deformation and movement [202]. For example, Ramanujan and Lao developed a ferrogel with magnetoelastic behavior based on Fe_3_O_4_ nanoparticles dispersed in PVA hydrogels for artificial muscle or soft actuator applications. The performance of this ferrogel to deflect in the presence of a magnetic field was evaluated, showing the strong dependence on Fe_3_O_4_ content. They demonstrated the design of a material able to mimic a finger-like motion under an external magnetic field achieved by incorporating magnetic nanoparticles [205].

Several studies for synthesizing smart hydrogels containing one-dimensional additives, including carbon nanotubes (CNTs), nanofibers, and nanorods, have been proposed. Zhang et al. reported the synthesis of PNIPAAm hydrogel prepared via incorporating single-walled carbon nanotubes (SWCNTs). The SWCNTs presence promoted an improvement in the thermal response time due to an enhanced water mass transport through the hydrogel, due to the generation of a large number of porous structures by the SWCNTs [206]. Moreover, the strong absorption of nanotubes allowed a material with a near-infrared optical response under a laser excitation of 785 nm. Therefore, this study demonstrated the development of thermally and optically responsive actuators with outstanding properties for smart solar device tracking systems or tissue connectors for biological media [206].

Combining 2D materials (such as nanosheets, nanoplates, and graphene oxides (GOs)) with hydrogels networks can also confer enhanced properties for the development of efficient actuators. The incorporation of anisotropic molybdenum disulfide (MoS_2_) nanosheets into PNIPAAm hydrogel with a tunable volume phase transition temperature (VPTT) was evaluated by Lei et al. [207]. The resulting actuator exhibited a dual response when exposed to heat or light sources. The anisotropic structure of the hydrogel led to a self-wrapping behavior capable of bending under manipulation controlled by light or heat. Their characteristics to undergo dual-responsive behaviors and mechanical performance make this material an interesting actuator with a promising future in soft robots.

Despite the several advantages of hydrogel-based actuators, these materials are designed to work only in an aqueous media, limiting their use in other possible environments. For example, in air conditions, it is a big challenge to achieve a fast response and forward motion of hydrogels [208].

### 5.4. Biosensors

Numerous studies have been conducted to build sensors from responsive hydrogels. Biocompatibility and other specific characteristics are required to fulfill the biosensor design and construction criteria in the biomedicine field. The innate behavior of smart hydrogels to respond to external stimulus, biocompatibility, and versatility, make it an excellent candidate for sensing applications.

Analyte/molecular recognition and high sensitivity properties are highly desired for sensors. The sensing mechanism is based on the biological recognition of a target analyte, which induces a volumetric change. Significant advances for employing smart hydrogel-based biosensors have been reported to treat and manage diseases by detecting biomolecules such as glucose, nucleic acids, proteins, and enzymes. In a study conducted by Zhang and coworkers, a multifunctional zwitterionic hydrogel was developed simultaneously to monitor pH and glucose levels in diabetic wounds. For this research, pH was controlled by adding phenol red dye sensitive in a specific pH range, whereas, for glucose detection, two sensing enzymes (glucose oxidase (GOx) and horseradish peroxidase (HRP)) were embedded into the poly-carboxybetaine (PCB) hydrogel. These two sensing enzymes catalyze glucose oxidation generating a dichlorofluorescein (DCF) fluorescent product. Moreover, an improvement in the activity and stability of artificial wound exudate was achieved. In vivo analysis was monitored, obtaining outstanding results in pro-healing ability compared to Duoderm, a commercial product employed for diabetic treatments [209].

Zhai et al. proposed one strategy for synthesizing a susceptible glucose enzyme sensor based on Pt nanoparticles (PtNPs)s homogeneously dispersed onto the polyaniline hydrogel (PANi). This sensor combined the properties of the PANi matrix with its conducting properties and the Pt nanoparticles as an active catalyst for the electro-oxidation of hydrogen peroxide, a by-product from GOx catalyzed reaction. Additionally, glucose oxidase (GOx) was also added for glucose recognition. The obtained result from the glucose enzyme sensor based on the PtNP/PAni hydrogel heterostructures showed an unprecedented sensitivity of 96.1 μA·mM^−1^·cm^−2^, with an average response time of three seconds, a linear range from 0.01 to 8 mM, and a low detection limit of 0.7 μM [210].

Additionally, other design strategies have focused on developing a wide variety of hydrogels sensors from different stimuli-responsive hydrogels. However, the incompatibility between materials represents a significant drawback for the design and integration of sensing materials and entities into hydrogel matrix [211].

## 6. Conclusions

Hydrogels are polymeric materials that have a wide range of applications, depending on the synthetic process used in their creation. In the synthesis process, certain properties can be conferred on the material such as biocompatibility, biodegradability, improved mechanical properties, and high thermal and chemical resistance, which can be used depending on its application. Besides, the cross-linked structure of the hydrogel can be achieved depending on the synthesis type or the raw materials employed. The physical hydrogels present reversible networks due to the weakness of their interaction. In contrast, the chemical hydrogels present permanent networks, although a minimum degree of swelling. The synthesis of both kinds of hydrogels has been significantly reported because each of them shows favorable characteristics. For example, as the cross-linking network increases or gains more strength, the swelling property will be reduced. Moreover, the hydrogels’ ability to combine with other structures such as nanoparticles to create multifunctional systems deserves special attention to enable successful applications.

Smart hydrogels arise as outstanding materials that can produce internal changes as a response to external changes. Smart hydrogels and their stimulus response have generated enormous interest in several fields, especially biomedicine, because of their biocompatibility and bio-degradability. These hydrogels have found exciting and potential applications such as the release of drugs in specific sites, tissue replacement, conversion of external energy into motion, and detection of specific biomolecules, among others. The research of stimuli-sensitive hydrogels will develop new promising outcomes in several fields during the following years.

## Figures and Tables

**Figure 1 gels-07-00182-f001:**
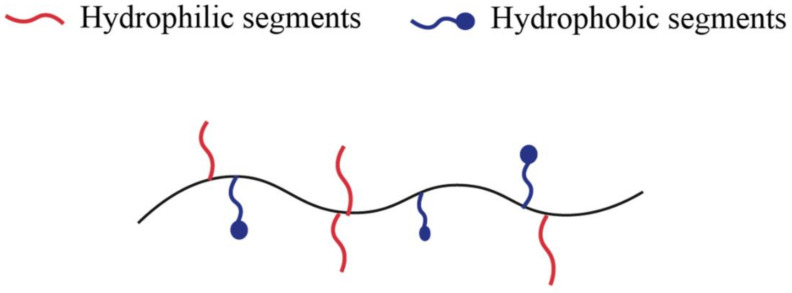
Schematic illustration of amphiphilic polymer.

**Figure 2 gels-07-00182-f002:**
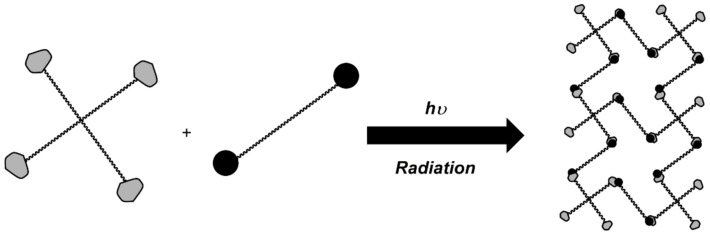
Representation of graft polymerization.

**Figure 3 gels-07-00182-f003:**
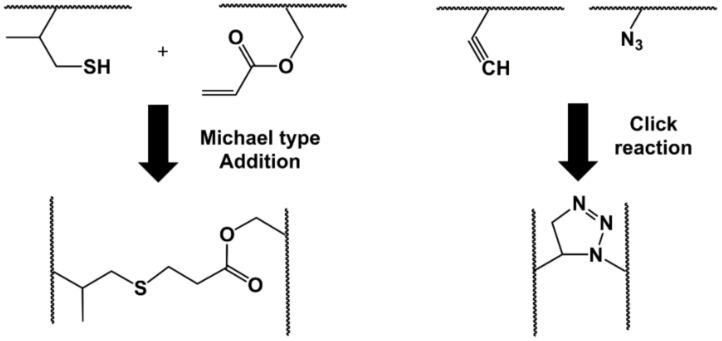
Representation of Michael-type addition and Click reactions forming a cross-linked structure.

**Figure 4 gels-07-00182-f004:**
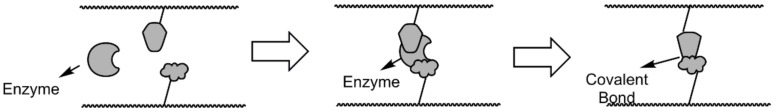
Schematic illustration of enzymatic method.

**Figure 5 gels-07-00182-f005:**
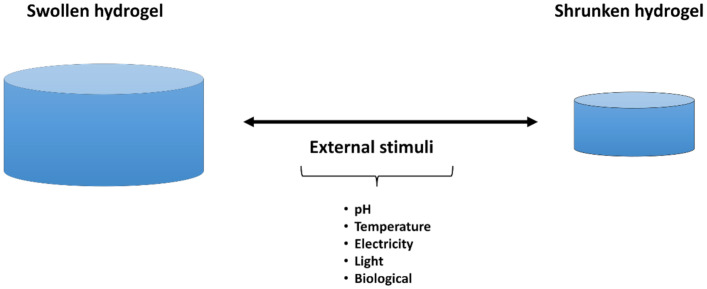
Schematic illustration smart hydrogels subjected to an external stimuli.

**Figure 6 gels-07-00182-f006:**
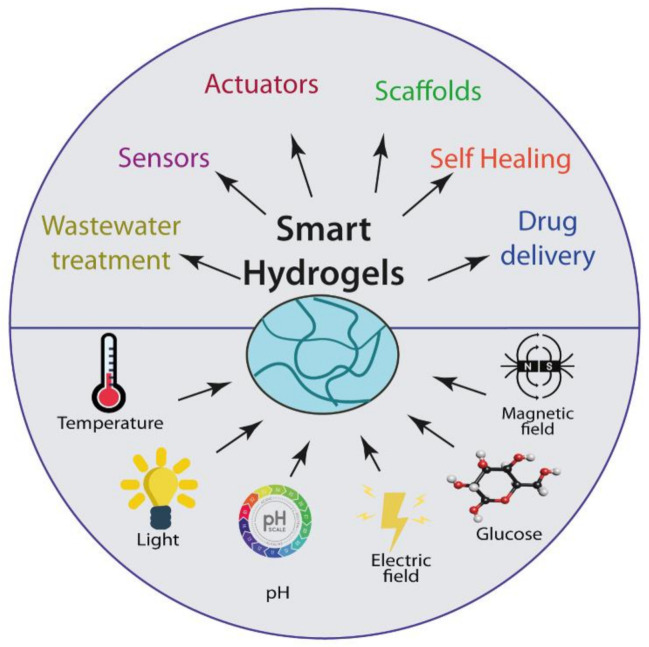
Stimulus-responsive hydrogels and their emerging applications.

**Figure 7 gels-07-00182-f007:**
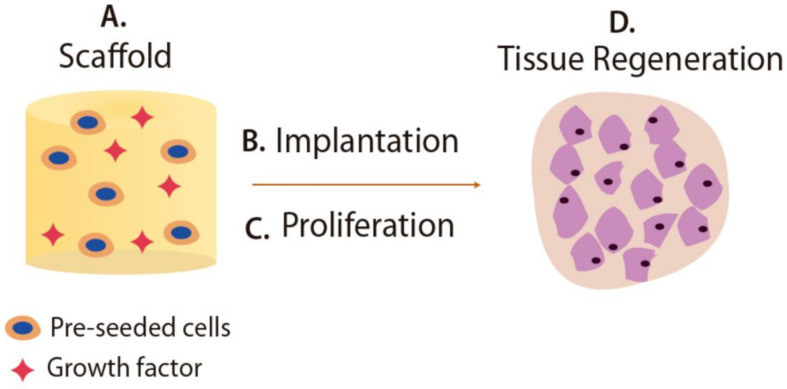
Schematic representation of the stages of scaffold-based tissue regeneration.

## Data Availability

Not applicable.

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
