# Peer review of "Hydrogels Classification According to the Physical or Chemical Interactions and as Stimuli-Sensitive Materials"

_gels, 2021, doi:10.3390/gels7040182_

Round 1
Reviewer 1 Report
The review presented here focuses on the properties and types of responsive hydrogels with various application areas. Mostly, internal structure interactions and their external responses were discussed in the review. Both physical and chemical changes in the internal sturucture of hydrogels were examplified with several literature studies. The categorization of hydrogels based on their responsiveness and applications seem to be organized.
- The titles for section 2 and 3 are probably ment to refer to crosslinking strategies, so it would be better to add „crosslinking stratgy“ term into titles. Otherwise „physical hydrogel“ title seems to be misleading.
- Figure 7 shows thermoresponsive LCST response of pNIPAM hydrogels in Drug Delivery Section! However in Section 2, temperature responsive hydrogels were mentioned, Figure 7 has nothing related to derug release or some other profile like that. That may create a controversial.
- The review gives the message of how advantageous hydrogel systems are due to their high water uptake capacity and swelling ability. However, especially dor drug delivery systems and tissue engineering applications, literature studies pointing to this feature seems to be less.
- Hydrogels are soft materials, but based on crosslinking strategies their mechanical properties vary, which contributes to their wide range of application areas. Althoug physical and chemical crosslinking startegies were discussed in section 2 and 3, the literature studies were not compared with each other about how these strategies change the mechnaical porperties of hydrogels?
Author Response
Reviewer 1.
The review presented here focuses on the properties and types of responsive hydrogels with various application areas. Mostly, internal structure interactions and their external responses were discussed in the review. Both physical and chemical changes in the internal sturucture of hydrogels were examplified with several literature studies. The categorization of hydrogels based on their responsiveness and applications seem to be organized.
- The titles for section 2 and 3 are probably ment to refer to crosslinking strategies, so it would be better to add „crosslinking stratgy“ term into titles. Otherwise „physical hydrogel“ title seems to be misleading.
Answer: Thanks for your suggestion, the titles were modified according to your recommendation.
- Figure 7 shows thermoresponsive LCST response of pNIPAM hydrogels in Drug Delivery Section! However in Section 2, temperature responsive hydrogels were mentioned, Figure 7 has nothing related to derug release or some other profile like that. That may create a controversial.
Answer: Thanks for your observations. The Figure 7 has been eliminated to avoid any controversy.
- The review gives the message of how advantageous hydrogel systems are due to their high water uptake capacity and swelling ability. However, especially dor drug delivery systems and tissue engineering applications, literature studies pointing to this feature seems to be less.
Answer: Thanks for your valuable comment. In the case of drug delivery, several authors highlight the swelling ability of hydrogels as a fundamental property that should influence the drug release rates, and the dissolution of the active drug. In addition, drug administration can involve different release mechanisms of the encapsulated drug in hydrogels. In delivery systems controlled by swelling/deswelling, the higher the swelling rate of the hydrogel, the greater the rate of drug release. Therefore, in this case, the rate and capacity of water absorption of the hydrogels play an important role in delivery systems.
- Hydrogels are soft materials, but based on crosslinking strategies their mechanical properties vary, which contributes to their wide range of application areas. Althoug physical and chemical crosslinking startegies were discussed in section 2 and 3, the literature studies were not compared with each other about how these strategies change the mechnaical porperties of hydrogels?
Answer: Thanks for your comment. A Table was added up by containing comparative information between physical and chemical cross-linking advantages.

Reviewer 2 Report
The paper is a review on hydrogels with emphasis on their biomedical applications. There are many reviews on this field. Hence, the most important feature is that the references in the paper are up dated. This seems to be the case. However, before publication I recommend a thorough revision of the english. There are many errors in the text. The author should also re consider the scientic part. Just a few examples:
Beginning of page 2. "In a hydrogel syntheses, a monomer and an initiator are necesssary". This is not true. It is possible to start with polymers (macromer) and to cross link these polymer chains already present. Also there are other types of cross-linking reactions besides processes involving radicals and initiators.
Middle of page 6. "Charges are ionized"???
Author Response
Reviewer 2.
The paper is a review on hydrogels with emphasis on their biomedical applications. There are many reviews on this field. Hence, the most important feature is that the references in the paper are up dated. This seems to be the case. However, before publication I recommend a thorough revision of the english.
Answer: Thank you for your recommendation, the English grammar has been revised and improved, as you can see in the document.
There are many errors in the text. The author should also re consider the scientic part. Just a few examples:
Beginning of page 2. "In a hydrogel syntheses, a monomer and an initiator are necesssary". This is not true. It is possible to start with polymers (macromer) and to cross link these polymer chains already present.
Answer: Thank you for your observations. Actually the hydrogel can be obtained from monomer, initiator and even by long-chain or short-chain polymers. These specific part has been corrected as you can see in the manuscript.
Also there are other types of cross-linking reactions besides processes involving radicals and initiators.
Answer: We greatly appreciate your comment. Our review is based on several articles and bibliographic reviews where chemical cross-linking is analyzed, involving both the presence of radicals in the hydrogel formation process and of crosslinking agents to ensure a successful synthesis. These methods ensure the formation of covalent bonds between the polymer chains. We add some references on this topic.
Arora, B., Tandon, R., Attri, P., & Bhatia, R. (2017). Chemical Crosslinking: Role in Protein and Peptide Science. Current Protein & Peptide Science, 18(9). https://doi.org/10.2174/1389203717666160724202806
Ermis, M., Calamak, S., Calibasi Kocal, G., Guven, S., Durmus, N. G., Rizvi, I., Hasan, T., Hasirci, N., Hasirci, V., & Demirci, U. (2018). Hydrogels as a New Platform to Recapitulate the Tumor Microenvironment. In Handbook of Nanomaterials for Cancer Theranostics(pp. 463–494). Elsevier. https://doi.org/10.1016/B978-0-12-813339-2.00015-3
Parhi, R. (2017). Cross-Linked Hydrogel for Pharmaceutical Applications: A Review. Advanced Pharmaceutical Bulletin, 7(4), 515–530. https://doi.org/10.15171/apb.2017.064
Wang, M., Guo, L., & Sun, H. (2019). Manufacture of Biomaterials. In Encyclopedia of Biomedical Engineering(pp. 116–134). Elsevier. https://doi.org/10.1016/B978-0-12-801238-3.11027-X
Middle of page 6. "Charges are ionized"???
Answer: Thanks for your observation,there was a mistake, which it has been corrected, as you can see in the manuscript.

Round 2
Reviewer 2 Report
The paper can be accepted for publication after further refinement of the english.